
# Sources of Springtime Surface Black Carbon in the Arctic: An Adjoint Analysis

Ling Qi[1,2], Qinbin Li[1,2], Daven K. Henze[3], Hsien-Liang Tseng[1,2] Cenlin He[1,2]

1. Department of Atmospheric and Oceanic Sciences, University of California, Los Angeles, CA, USA

2. Joint Institute for Regional Earth System Science and Engineering, University of California, Los Angeles, CA, USA

3. Department of Mechanical Engineering, University of Colorado, Boulder, CO, USA

**Abstract:** We quantify source contributions to springtime (April 2008) surface black carbon (BC) in the Arctic by interpreting surface observations of BC at five receptor sites (Denali, Barrow, Alert, Zeppelin, and Summit) using a global

chemical transport model (GEOS-Chem) and its adjoint. Contributions to BC at Barrow, Alert, and Zeppelin are dominated by Asian anthropogenic sources (40–43%) before April 18 and by Siberian open biomass burning emissions (29–41%) afterward. In contrast, Summit, a mostly free tropospheric site, has predominantly an Asian anthropogenic source contribution (24–68%, with an average of 45%). We compute the adjoint sensitivity of BC concentrations at the five sites during a pollution episode (April 20−25) to global emissions from March 1 to April 25. The associated contributions are the

combined results of these sensitivities and BC emissions. Local and regional anthropogenic sources in Alaska are the largest anthropogenic sources of BC at Denali (63%), and natural gas flaring emissions in the Western Extreme North of Russia (WENR) are the largest anthropogenic sources of BC at Zeppelin (26%) and Alert (13%). We find that long-range transport of emissions from Beijing-Tianjin-Hebei (also known as Jing-Jin-Ji), the biggest urbanized region in Northern China, contribute significantly (~10%) to surface BC across the Arctic. On average it takes ~12 days for Asian anthropogenic

emissions and Siberian biomass burning emissions to reach Arctic lower troposphere, supporting earlier studies. Natural gas flaring emissions from the WENR reach Zeppelin in about a week. We find that episodic, direct transport events dominate BC at Denali (87%), a site outside the Arctic front, a strong transport barrier. The relative contribution of direct transport to surface BC within the Arctic front is much smaller (~50% at Barrow and Zeppelin and ~10% at Alert). The large contributions from Asian anthropogenic sources are predominately in the form of 'chronic' pollution (~40% at Barrow and

65% at Alert and 57% at Zeppelin) on 1–2 month timescales. As such, it is likely that previous studies using 5- or 10-day trajectory analyses strongly underestimated the contribution from Asia to surface BC in the Arctic. Both finer temporal resolution of biomass burning emissions and accounting for the Wegener-Bergeron-Findeisen (WBF) process in wet scavenging improve the source attribution estimates.

## 1. Introduction

The Arctic, one of the most sensitive regions to climate change, warms at a rate twice as rapid as the global average (AMAP, 2011). Climate modeling studies indicate that the Arctic surface warms from global BC, which absorbs solar radiation in the





atmosphere (Bond et al., 2013 and references therein). Specifically, atmospheric BC at lower latitudes warms the Arctic surface due to polar-ward transport of heat absorbed by BC (0.7–0.8 K (W m$^{-2}$)$^{-1}$, Shindell and Faluvegi, 2009; Sand et al. 2013, 2015). Although the total effect of BC in the Arctic troposphere is warming the surface, BC at different altitudes has different climate effects on the Arctic surface (Shindell and Faluvegi, 2009; Flanner et al., 2013). BC in the upper

tropospheric Arctic slightly cools (-0.2 ± 0.1 K (W m$^{-2}$)$^{-1}$) the Arctic surface due to surface dimming, while surface atmospheric BC exerts a strong surface warming effect (2.8 ± 0.5 K (W m$^{-2}$)$^{-1}$, Flanner, 2013). In addition, this climate effect has a strong seasonal variation with the largest mass-normalized Arctic warming in spring, when the high insolation and surface albedo strongly facilitate a large radiative forcing of BC and the associated surface warming (Quinn et al., 2008; Flanner, 2013). It is thus imperative to improve our understanding of the distribution of BC in springtime in the Arctic and

identify the sources both in the troposphere and at surface.

Numerous studies have analyzed BC vertical profiles and identified sources of springtime BC in the troposphere in the Arctic (e.g. Liu et al., 2015 and references therein), particularly during polar year 2008. They found that a dense haze layer with maximum BC concentration is usually observed in the middle troposphere (Warneke 2009, 2010; Wang et al., 2011;

Brock et al., 2011; Marelle et al., 2015) and their sources are largely diverse. Stohl et al. (2007) found that agricultural fires in Eastern Europe strongly enhanced BC concentration in the European Arctic in spring 2006. Studies based on aircraft observations from Arctic Research of the Composition of the Troposphere from Aircraft and Satellites (ARCTAS, Jacob et al., 2010), and Aerosol, Radiation, and Cloud Processes affecting Arctic Climate (ARCPAC, Brock et al., 2011) found that agricultural as well as boreal forest fires in south Siberia dominated BC concentrations (up to 80%) in a large part of the

Arctic troposphere (particularly in North American sector) in spring 2008 (Warneke et al., 2009, 2010; Brock et al., 2011; Wang et al., 2011). Given that an increasing trend of frequency of boreal forest fires has been predicted due to global warming, the contribution of these fires to Arctic BC is likely to increase in the future (Soja et al., 2007; Flannigan et al., 2009; Wotton et al., 2010; Liu et al., 2012). With the rapid increase of anthropogenic emissions in Eastern and Northern Asia, recent studies have shown that this region may become an important source (up to 90%) of Arctic BC in late winter

and early spring, particularly in the free troposphere (Shaw et al., 2010; Frossard et al., 2011; Liu et al., 2015). Other studies suggest otherwise because strong scavenging during uplift processes in warm conveyer belts can significantly decrease the BC transport efficiency and the resulting contribution from this source (Stohl, 2006; Matsui et al., 2011).

The Arctic surface is isolated from the free troposphere by a strong transport barrier, the so-called 'Arctic front', which is a

closed dome formed by surfaces of constant potential temperature (Stohl, 2006 and references therein). As such, the Arctic front strongly reduces the influence from episodic pollution entering the Arctic in the middle and upper troposphere and limits their impact on surface BC concentrations (Stohl, 2006; Brock et al., 2011). In addition, emissions in the Arctic front





are trapped in the lower troposphere and enhance surface concentration. The Arctic front extends further south (40°N) over Europe and Russia in winter and early spring (Stohl, 2006). Thus, sources of Arctic surface BC have been repeatedly identified as emissions from the two regions. A large fraction of the 5- or 10-day back trajectories ends in the industrial regions in Russia and Europe (Pollisar et al., 2001; Sharma et al., 2006; Huang et al., 2010; Dutkiewicz et al., 2014). In

addition, long-term observations (decades) at surface sites Barrow (Pollisar et al., 2001; Sharma et al., 2006; Hirdman et al., 2010), Alert (Sharma et al., 2004, 2006; Huang et al., 2010; Hirdman et al., 2010), Zeppelin (Eleftheriadis et al., 2009; Hirdman et al., 2010), and Kevo (Dutkiewicz et al., 2014) showed that BC concentration has steadily declined since 1980. They relate these declining trends with declining emissions in the former Soviet Union. Asian contributions to surface BC were considered not to be significant due to two reasons. First, very few back trajectories ended in this region (Pollisar et al.,

2001; Sharma et al., 2006; Huang et al., 2010; Dutkiewicz et al., 2014). Second, observed atmospheric surface BC concentrations at Alert, Barrow and Zeppelin did not increase with increasing BC emissions in East Asia since 2000 (Sharma et al., 2013). In addition, most of these studies were focused on the long-term average sources and contrasted sources in winter and in summer. Few studies have identified sources of BC in springtime surface Arctic, which efficiently warms the Arctic surface (Flanner, 2013).

Previous studies on BC source apportionment have used either statistical analysis of trajectories (Pollisar et al., 2001; Sharma et al., 2004; Eleftheriadis et al., 2009; Stohl, 2006; Hirdman et al., 2010; Harrigan et al., 2011, Dutkiewicz et al., 2014), the tagged tracer technique (Wang et al., 2011; Wang et al., 2014) or sensitivity simulations with perturbed emissions (Koch and Hansen, 2005; Shindell et al., 2008; Huang et al., 2010a; Bourgeois and Bey, 2011; Sharma et al., 2013; Ma et al.,

2013). Trajectory analysis efficiently identifies transport pathways, but the model integration error increases when modeling time exceeds 5–6 days. This error introduces large uncertainties to the back trajectory paths, thereby sources can only be reliably assigned on continental scales (Liu et al., 2015). In addition, the assumption of the trajectory method is that the tracer is inert and not affected by chemical or other removal processes, which introduce large uncertainties to source apportionment (Liu et al., 2015). Tagged tracer and sensitivity simulations include chemical and physical processing of BC

but are computationally limited in the spatial and temporal resolution of the source regions that can be considered (Wang et al., 2014). The Chemical Transport Model (CTM) adjoint not only explicitly simulates chemical and physical processes of aerosols but also offers a far more computationally efficient approach for receptor-oriented source attribution (Henze et al., 2007; Hakami et al., 2007). A single run of the adjoint model can compute the sensitivity of BC concentrations at a given location (or an average over a spatial domain) and time (or an average in a time interval) to global emissions over the spatial

and temporal resolution of the model (Zhang, L. et al., 2009). The method has been applied to examine transpacific (Zhang, L. et al., 2009) and Arctic (Walker et al., 2012) transport of ozone, source apportionment of aerosol pollution episodes





(Nester and Panitz, 2006; Zhang, L. et al., 2015a), and BC radiative forcing in the Himalayas and Tibetan Plateau (Kopacz et al., 2011).

In a previous study (Qi et al., 2016a), we assessed the sensitivity of BC concentration in surface air and in snow in the Arctic

to flaring emissions, dry deposition velocity over snow and ice, and WBF in mixed-phase clouds. With all the improvements, simulated BC concentrations in snow in eight Arctic sub-regions agree with observations within a factor of two, meanwhile, simulated surface atmospheric BC fall within the range of observations. Based on this improved simulation of BC distribution in the Arctic, we use here the tagged tracer technique in a global 3D chemical transport model, GEOS-Chem, to identify sources of BC at Arctic surface sites (Denali, Barrow, Alert, Zeppelin and Summit) resolved at continental scales in

April, 2008. We then use the GEOS-Chem adjoint to refine our estimated source contributions to BC at the five sites to sources resolved at the 2° latitude × 2.5° longitude horizontal scale and hourly temporal resolution during a pollution event (April 20–25).

## 2. Surface BC Observations

In-situ measurements of BC in April 2008 are available at five sites within the Arctic Circle (Fig. 1): Denali, AL (63.7°N,

149.0°W, 0.66 km a.s.l.) in the low Arctic, three sites in the high Arctic, Barrow, AL (71.3°N, 156.6°W, 11 m a.s.l.), Alert, Canada (82.3°N, 62.3°W, 210 m a.s.l.), and Zeppelin, Norway (79°N, 12°E, 478 m a.s.l.), and a free tropospheric site at Summit, Greenland (72.6°N, 38.5°W, 3.22 km a.s.l.). Denali, outside the Arctic front (~66°N), receives pollution transported to the Arctic via 'Aleution Storm Track' displaced to the central and north Alaska in April 2008 (Fuelberg et al., 2010). Barrow and Alert are within the Arctic front and experience regular temperature inversion (Sharma et al., 2013). This

inversion strongly suppresses the vertical transport from above and traps emissions in the Arctic front in the lower troposphere (Stohl, 2006; Brock et al., 2011). Barrow is on the Northern land tip of Alaska, 8 km northeast of the town Barrow. It is influenced by both marine and continental air (Hirdman et al., 2010). Alert is located the furthest north of all five sites and is most isolated from continental sources (Hirdman et al., 2010). The Zeppelin observatory is on a mountain ridge on the Svalbard archipelago. Because of the relatively high elevation (478 m), Zeppelin is not always in a stable

inversion layer (Sharma et al., 2013). Summit (3.2 km), on the top of the Greenland glacial ice sheet, is always in the Arctic free troposphere (Hirdman et al., 2010). BC is measured by Thermal Optical Reflectance combustion method at Denali, by a particle soot absorption photometer at Barrow, and by an aethalometer at Alert, Zeppelin, and Summit. Uncertainties of these measurements lie in overestimate of absorption based on light transmission, which is also affected by scattering. Additionally, empirical conversion from optical response to BC mass also involves uncertainties. Finally, there are

uncertainties from non-BC absorbers. Details are summarized in Qi et al. (2016a).



### 3. GEOS-Chem and its adjoint

### 3.1 GEOS-Chem simulation of BC

GEOS-Chem is a global CTM driven with assimilated meteorology from the Goddard Earth Observing System (GEOS) of the NASA Global Modeling and Assimilation Office (GMAO). We use GEOS-5 meteorological data set to drive model

simulation at 2° latitude × 2.5° longitude horizontal resolution and 47 vertical layers from the surface to 0.01 hPa. The lowest model levels are centered at approximately 60, 200, 300, 450, 600, 700, 850, 1000, 1150, 1300, 1450, 1600, 1800 m above sea level. Tracer advection, moist convection, deep convection and shallow convection schemes are summarized in Qi et al. (2016a).

Northern hemispheric BC emissions in April are shown in Fig. 2. Anthropogenic emissions of BC are from Bond et al. (2007) with Asian emissions from Zhang, Q. et al. (2009). We applied seasonal variation for domestic heating emissions, which are concentrated in winter at high latitude, based on the heating degree day concept (Stohl et al., 2013). Open biomass burning emissions are available from the Global Fire Emissions Database version 3.1 (GFEDv3) with updates for small fire emissions (Randerson et al., 2012). We use the GFED inventory with two temporal resolutions, the standard monthly

inventory and a 3-hourly inventory. We use the monthly inventory for the standard simulation and the 3-hourly inventory for the uncertainty simulation (Sect. 4.3.1). To derive the 3-hourly inventory, daily emissions are first re-sampled temporally from the monthly inventory according to Moderate Resolution Imaging Spectroradiometer (MODIS) daily active fire counts (Giglio et al., 2003). Then a diurnal cycle with a 3-hourly time step based on the active fire observations (Prins et al., 1998) is applied to the daily inventory. The resulting 3-hourly inventory has the same overall emissions as the monthly inventory

but with a much finer temporal distribution. We also include emissions from gas flares in the oil and natural gas industry (Stohl et al., 2013). It is suggested that the total emissions of gas flaring only account for 3% of global total BC emissions but they account for 42% of total BC emissions in the Arctic (Stohl et al., 2013). They are significant contributors to both BC deposition (~20%) and ambient BC concentrations during snow season (September to April) in the Arctic (Qi et al., 2016a). BC emissions in the Arctic are shown in Fig. 1.

We assume 80% of the freshly emitted BC particles are hydrophobic and become hydrophilic with an e-folding time of 1.15 days, which reproduces the Asian outflow (Park et al., 2003; 2005). Liu et al. (2011) implemented an OH-dependent aging scheme, which has a strong diurnal and seasonal variation. Slower aging in winter allows more BC to remain hydrophobic and be transported to the Arctic, resulting in a better comparison with surface observations. The change in other seasons is

30 insignificant. We implement this aging scheme in GEOS-Chem and also find insignificant change of tropospheric and surface BC concentration in the Arctic in April (see supplement material Fig.1S). We estimates dry deposition velocity of



BC over snow and ice using resistance-in-series method (Wesely 1989; Zhang et al., 2001), validated by recent

measurements of aerosol deposition velocity over snow and ice (Qi et al., 2016a). Wet scavenging of BC follows Wang et

al., (2011) with updates for BC scavenging efficiency in mixed-phase clouds (Qi et al., 2016b). We parameterize BC

scavenging efficiency in mixed-phase clouds accounting for the effects of the WBF process (Qi et al., 2016b). WBF occurs

when environmental vapor pressure is above the saturation vapor pressure of ice crystals and below that of cold water drops.

Ice crystals grow and cold water drops evaporate, releasing BC particles in the cold water drops back into interstitial air. This

process strongly reduces BC scavenging efficiency globally (from 8% in the tropics to 76% in the Arctic), slows down wet

scavenging and increases atmospheric BC concentrations. Including WBF significantly improves the simulation of BC

distribution in air (discrepancy reduced from -65% to 30%) and in snow (discrepancy is halved both in mid-latitudes – from

34% to 17% and in the Arctic – from -20% to -10%) globally (Qi et al., 2016b). We tested the uncertainties of source

attribution associated with WBF in Sect 4.3.2.

We identify sources of BC in spring Arctic using the tagged tracer technique, which is a physically consistent and

computationally efficient approach to attribute sources resolved at continental scales (Wang et al., 2011; Wang et al., 2014).

BC emitted from different source types (anthropogenic versus open biomass burning emissions) and source regions (Europe,

Russia, Asia and North America) are tagged with no overlap among these geographical regions (Fig. 2). BC emitted in these

tagged regions are explicitly tracked in the model and are treated the same way (transport, chemical and physical processes)

as the original BC, allowing for direct estimate of the contributions from individual tagged source types and source regions.

We spin up the model by two months before the starting date of March 1.

**3.2 GEOS-Chem adjoint simulation of BC**

Compared with the tagged tracer technique, the adjoint modeling approach computes source-receptor sensitivities for

individual receptor locations more efficiently and can be done at a much finer temporal and spatial resolution. We use

GEOS-Chem adjoint (Henze et al., 2007) model v35 with updated emissions, dry deposition velocity and wet scavenging as

described in Sect. 3.1. The adjoint model has previously been used to constrain emissions of CO (Kopacz et al., 2009, 2010),

BC (Mao et al., 2014; Zhang et al., 2015) and other aerosols (Henze et al., 2009) and to identify sources of ozone (Zhang et

al., 2009), BC (Kopacz et al., 2011) and other aerosols (Zhang et al., 2015). In addition, the adjoint has also been used to

estimate the sensitivity of direct radiative forcing to aerosol emissions (Henze et al., 2012).

In this study, we use the adjoint to compute the sensitivity of BC concentrations ([BC]$_s$) at the five receptor sites in the

Arctic to global BC emissions ($e$) resolved at 2° latitude × 2.5° longitude horizontal resolution over the history of air parcels

reaching the sites (March 1–April 25). This sensitivity is denoted as $k$,





$$[BC]_s = \boldsymbol{k} \times \boldsymbol{e} \qquad (1)$$

As horizontal advection in GEOS-Chem is approximately linear, and the rest of the chemical and physical process are entirely linear, multiplication of the sensitivity ($\boldsymbol{k}$) by emissions ($\boldsymbol{e}$) yields an estimate of how much BC emissions from each grid cell contribute to BC concentrations at a receptor site ($[BC]_s$) (Henze et al., 2007; Kopacz et al., 2011). In this study,

5  $[BC]_s$ is the mean BC concentration at each receptor site (Fig. 1) during April 20−25, 2008. The sensitivity is propogated backwards in time from April 25 to March 1. Then, hourly contributions from each grid cell can be calculated by multiplying these sensitivities with emissions at that time. Integrating these contributions from March 1 to April 25 gives the contributions of emissions in each grid cell to $[BC]_s$ at the receptor sites. Further integrating these contributions globally approximates the mean $[BC]_s$ during April 20–25 at each site.

We validate GEOS-Chem adjoint via comparison of adjoint gradients to forward model sensitivities, Λ, calculated using the finite difference approximation,

$$\Lambda = \frac{J(\sigma + \delta\sigma) - J(\sigma)}{\delta\sigma} \qquad (2)$$

where $J$ is the cost function, defined as the mean $[BC]_s$ at each of the five receptor sites, $\sigma$ the scaling factor for BC

emissions (σ = 1 for the sensitivity simulation). We use $\delta\sigma = 0.1$ for all tests in this study. We compare the sensitivities of mean $[BC]_s$ in April 20–25 at each receptor site to anthropogenic and biomass burning emissions in March 1–April 25 estimated by finite difference and adjoint method. This evaluation is quite time consuming to perform at each grid cell throughout the globe owing to the expense of the finite difference calculations ($144 \times 91$ runs). Hence, we selected 10 model grid cells with the largest anthropogenic and open biomass burning sources for the validation, because the absolute

difference between the two methods is more substantial for larger values (Henze et al., 2007).

Fig. 3 shows the adjoint validation results. The simulation is for March 1–April 25 and the cost function $J$ is evaluated at the end of each simulation. The agreements between finite difference sensitivities and the adjoint gradients are within ~15% (slopes vary from 0.84 to 1.15, except for biomass burning contribution at Summit), largely within the uncertainty that arises

from deriving the adjoint of the advection using the continuous approach despite their being nonlinearities in the discrete treatment of advection in the forward model (Henze et al., 2007). To quantify the discrepancy, we compare the sensitivities estimated by finite difference and the adjoint method at one grid cell in biomass burning regions in Siberia, with advection turned on and off. Without advection, finite difference sensitivities agree with adjoint gradients to within 1%. However, with advection turned on, the difference increases to ~15%. Long-range transport from source regions to receptor sites in the

Arctic does not exaggerate the disagreement. In addition, simulation lengths of 1, 5, 10, 20 and 50 days are not found to substantially alter the overall comparison, indicating that this error does not accumulate in time. In addition, the adjoint

gradients estimated by the continuous advection scheme are likely smoother and more physically meaningful than the finite difference sensitivities estimated by the discrete advection scheme (Henze et al., 2007; Liu et al., 2008; Gou et al., 2011). For example, a negative value of finite difference sensitivity is shown for anthropogenic sources for Denali and an abnormally low finite difference sensitivity value is also shown for anthropogenic sources for Barrow, because of the

discrete advection scheme used in the forward model (Thuburn and Haine, 2001; Henze et al., 2007; Hakami et al., 2007).

## 4. Results and Discussion

### 4.1 Sources of Arctic BC in April 2008

#### 4.1.1 Sources contributing to BC at selected surface sites

Fig. 4 (left panels) shows measured and GEOS-Chem simulated daily mean BC concentrations at the five sites for April

2008. At Denali, the model reproduces both the monthly mean (within 26%) and the day-to-day variation (correlation coefficient $r = 0.96$). At Barrow and Alert, simulated monthly mean BC concentrations agree with observations to within 10%. At Zeppelin, the monthly mean BC is overestimated by 80%. This positive bias is likely because of two reasons. First, GEOS-Chem does not reproduce the spatial and temporal variation of flaring emissions in WENF, which is one of the major sources of BC at Zeppelin (Stohl et al., 2013; Qi et al., 2016a). Second, the model does not properly differentiate WBF-

versus riming-dominated in-cloud scavenging in mixed-phase clouds (Qi et al., 2016a). At Summit, a free tropospheric site, monthly mean BC concentration is overestimated by 60%.

Using tagged tracers, we compute the contribution from major continental-scale source regions (Fig. 2) to surface BC in the Arctic (Fig. 4, right panels). Asian anthropogenic sources make the largest contribution to surface BC (e.g., 32−35% at

Barrow, Alert and Zeppelin) and free tropospheric BC (45% at Summit) in the Arctic. Other large anthropogenic sources are from North America (at Denali and Barrow) and Siberia (at Zeppelin) because of their close proximity. Anthropogenic contributions from Europe, Siberia and North America to BC at Alert are comparable.

We also find that Asian anthropogenic contribution sharply increase (double or triple) from January–March (not shown) to

April across the five sites, although BC emissions in Asia are much lower in April than in the previous three months (by 25%) due to less energy consumption from domestic heating. In contrast, contributions from European, North American, and Siberian anthropogenic sources are relatively flat from January–March to April. Such contrast indicates that the pole-ward transport of Asian emissions in April is enhanced so much so that it offsets the relatively lower emissions in that month. Previous studies suggest that Asian emissions are probably underestimated (Fu et al., 2012; Zhang et al., 2015) by about a

factor of two. If the actual Asian emissions were higher, Asian contribution to the Arctic is likely larger than estimates in this





study. Additionally, given that Asian BC emission is likely to continue to rise over the coming years (Qin and Xie, 2011; Wang et al., 2012), it is likely that their contribution to springtime BC in the Arctic will likewise increase.

Observations show a strong enhancement of BC concentration at Denali (up to ~400 ng m$^{-3}$) during April 18−30 (Fig. 4, top left). GEOS-Chem reproduces this strong transport event (to within 20%). This enhancement is from a strong increase of contribution from North American anthropogenic sources (up to ~200 ng m$^{-3}$) and Siberian biomass burning emissions (up to ~200 ng m$^{-3}$) (Fig. 4, top right) due to favorable meteorological conditions over the North Pacific for pole-ward transport (Sect. 4.2.1). The Siberian biomass burning emissions are from forest fires in southern Siberia-Lake Baikal area and agricultural burning in Kazakhstan-southern Russia (Warneke 2009, 2010; Wang et al., 2011). These biomass burning emissions also enhance BC concentrations at the other sites, although with much smaller magnitudes (up to ~60 ng m$^{-3}$) and at different times. At Denali, the contribution of Siberian biomass burning reaches its maximum in April 18–25. At Barrow, two maxima appear at April 18 and 24. At Alert and Summit, the Siberian biomass burning contributions are the largest during April 25–30. At Zeppelin, the two peaks are at April 22 and 30. The relative contributions from biomass burning emissions to BC at the five sites reach up to 46−64%, exceeding the contribution from anthropogenic sources during the pollution event. Yttri et al. (2014) derived BC concentration from biomass burning at Zeppelin based on the ratio of Levoglucosan and BC in 2008. They found that the lower and upper estimates of biomass burning contributed BC at Zeppelin were 4.3−13.3% and 31−45%. This estimate is broadly consistent with the range from this study (5.2−55.1% with a mean of 17.6%). It indicates that ways to mitigate open biomass burning can be effective at reducing springtime surface BC in the Arctic and thus lessen the BC snow albedo effect (Flanner, 2013).

### 4.1.2 Atmospheric lifetimes of Arctic BC

Using tagged tracers (Sect. 3.1), we show GEOS-Chem simulated transport pathways of Arctic BC in April 2008 from major sources (Fig. 2) in Figs. S2 and S3. These transport pathways are in broad agreements with previously identified pathways of BC reaching the Arctic surface (Hirdman et al., 2010; Dutkiewicz et al., 2014) and the troposphere (e.g. Klonecki et al., 2003; Stohl, 2006; Wang et al., 2014). For instance, anthropogenic emissions from Siberia and Europe are transported to the Arctic through low-level transport (Klonecki et al., 2003; Stohl, 2006; Harrigan et al., 2011; Marelle et al., 2015), while anthropogenic emissions from Asia and biomass burning emissions from South Asia are uplifted in source regions and enter the Arctic through the middle and upper troposphere (Matsui et al., 2011; Wang et al., 2014; Liu et al., 2015). These transport pathways result in considerably different lifetimes of Arctic BC against deposition (Table 1). For example, BC from Europe and Siberia enters the Arctic through the lower troposphere where it experiences relatively fast dry and wet deposition (Bourgeois and Bey, 2011; Wang et al., 2014). Consequently, the lifetimes are relatively short (7−9 days annually, 10−16 days for March−April). In contrast, BC from Asian anthropogenic and South Asian biomass burning





emissions is transported into the Arctic middle and upper troposphere where deposition is relatively weak. As expected, the resulting lifetimes are much longer (~1.5–3 months annually and ~2–4 months for March–April). The lifetimes of BC from North American anthropogenic sources and Siberian biomass burning sources fall somewhere in between (2–3 weeks annually and ~1 month for March–April). Stohl (2006) estimated that the Arctic age of particles against transport decrease

with increasing height (surface: up to 2 weeks; 3–5 km: ~5 days; 5–8 km: ~3 days). Thus, BC lifetime in the Arctic is determined by deposition and transport at the surface and is dominated by transport in the middle and upper troposphere. If both deposition and transport were considered, BC particles in the middle troposphere might have the longest lifetime in the Arctic.

### 4.2 Adjoint source attribution of Arctic BC during April 20–25, 2008

Tagged tracers can identify sources of BC efficiently at large geographical (e.g., continental) scales (see Sect. 4.1). However, for episodic transport, source attribution at much finer spatial and temporal resolutions is often needed. In this section, we focus our analysis on the strong pollution event during April 20−25 most notably at Denali (Fig. 4). We use the GEOS-Chem adjoint to estimate sources of BC at all five sites at 2° latitude × 2.5° longitude horizontal and hourly temporal resolution.

### 4.2.1 Sensitivity of surface Arctic BC to the Northern Hemispheric emissions

Fig. 5 shows the sensitivity of mean BC in April 20−25 at the five surface sites to emissions in the Northern Hemisphere during the last 5, 10 and 25 days (dates back from April 25) of transport. The sensitivities are computed per model grid cell using results from the adjoint simulations (Sect. 3.2) and integrated in the given time periods (5, 10 or 25 days). BC at the four surface sites is most sensitive to nearby sources within 5 to 10 days prior to reaching these sites. When integrated over the 25 days prior to arrival, Denali, outside the Arctic front, is most exposed to emissions in the Bering Sea and Russia.

Barrow, Alert and Zeppelin in the high Arctic are most sensitive to emissions in the Arctic front. More specifically, BC at Barrow, Alert and Zeppelin is most sensitive to emissions in North America, the North Pole and Europe, respectively. Our estimates are consistent with sensitivity patterns of the transport climatologies (2000-07) in winter from Hirdman et al. (2010) for the three sites. The sensitivity of BC at Summit is clearly different from the other four low-level surface sites. BC at Summit in the free troposphere is more sensitive to emissions at lower than at higher latitudes, consistent with a previous

study (Hirdman et al., 2010). Because of its higher altitude, Summit (3.2 km), more than the other four Arctic surface sites, is frequently exposed to air parcels from warmer low-latitudes that rise isentropically and transported northwards (Hirdman et al., 2010). In addition, Summit is more sensitive to emissions from the United States but relatively less sensitive to Eurasian emissions than the other four surface sites. Emissions in the Arctic front do not influence BC concentration at Summit because the stable atmosphere in the Arctic front during the simulation period in this study strongly suppresses the vertical

transport of emissions from surface to the free troposphere. The difference of sensitivities for Summit between this study and




Hirdman et al. (2010) is that this study does not show a strong sensitivity to emissions over Greenland, where is the highest sensitivity shown in Hirdman et al. (2010). This difference is probably resulted from different seasons and years simulated in the two studies (this study: April 20–25, 2008; Hirdman et al. (2010): climatology (2000–07) in winter and summer).

**4.2.1 Source attribution of surface Arctic BC based on adjoint sensitivity**

Fig. 6 shows contributions from global BC emissions during March 1–April 25 to BC concentrations during April 20−25 at Denali, Barrow, Alert, Zeppelin and Summit as computed from the GEOS-Chem adjoint simulations (Sect. 3.2). The left panels show the contributions integrated from March 1 to April 25. The right panels are hourly contributions from major source regions (as defined in Fig. 2) during March 1–April 25. The left panels are BC concentrations originated from each model grid cell subsequently transported to the receptor sites. Summing these values up globally approximates (within 15%,

see Sect. 3.2) the mean BC concentrations in April 20–25 at the receptor sites. Summing up these values in a specific region (e.g. regions defined in Fig. 2), i.e., integrating the area underneath an individual curve (Fig. 6, right panels), gives the overall contribution from BC emitted in that source region during March 1–April 25. The results are summarized in Table 2. The difference between this adjoint attribution and the tagged tracer method is within 15% (Table 2). Such difference is largely explained by the choices of discrete (in the forward simulation) versus continuous (in the adjoint derivation)

treatment of advection (Sect. 3.2). Transport timescales from different sources to the five sites can also be inferred from the spectra in the right panels.

Previous studies found that boreal forest fires in the southern Siberia-Lake Baikal area (40–60°N, 100–140°E) and grass/crops burning in Kazakhstan-southern Russia (40–60°N, 30–90°E) were the major sources of BC sampled along the

20 ARCPAC and ARCTAS flights in April 2008 based on MODIS fire detection (Warneke et al., 2009, 2010). We aggregate separately the contributions from forest fires and grass/crops burning to BC at the five receptor sites. The resulting adjoint sensitivity-based estimates show that Siberian biomass burning contributions at the five sites are predominantly (> 90%) from forest fires in the southern Siberia-Lake Baikal area. Forest fires in that region during April 1−14 have large influences on BC at the two Alaskan sites Denali and Barrow, reaching these two sites after 11–25 days of transport (Fig. 6, right

panels). The contribution from those forest fires, when integrated during April 1–14, is four times larger at Denali (137.8 ng m$^{-3}$) than at Barrow (30.4 ng m$^{-3}$). Note the different $y$-axis ranges (Fig. 6, right panels). The contributions at Alert and Zeppelin show similar temporal distributions, with relatively large influences from emissions during April 1–6 and 7−12, but different intensities. The contribution is a factor of three higher at Zeppelin (39.3 ng m$^{-3}$) than at Alert (14.3 ng m$^{-3}$).

In contrast, global anthropogenic contributions are considerably more scattered. At Denali, for instance, local and regional emissions – power plants (such as North Pole, US EIA, 2010) and petroleum refinery industries near Fairbanks (US EIA,





2016) account for 63% of total global anthropogenic contribution to BC at Denali. Additionally, we find that the two sites are influenced by long-range transport of emissions from Northeast China, particularly residential emissions from Heilongjiang province. BC emissions from natural gas flares in the WENR are significant sources of BC in the Arctic (Stohl et al., 2013; Qi et al., 2016a). These flares are the largest sources to BC at Alert (13% of the global anthropogenic contributions) and Zeppelin (26% of global anthropogenic contributions). Global contributions to BC at Summit are distributed more evenly geographically compared to the other four sites. An interesting feature of the anthropogenic contributions is that industrial and residential BC emissions from the Jing-jin-ji cluster of megacities and Shandong province in East China (together they encompass six model grid cells) are common important sources (7–10% of global contributions) of BC at all five sites.

At Denali, and to a much lesser degree at Barrow, significant contributions are seen from Asian anthropogenic emissions of BC emitted during April 8–15, with a maximum contribution from emissions on April 11. These emissions arrived at the sites after 12–17 days of transport. This episode results from 'direct' transport from Asia to the two sites (Brock et al., 2011, and references therein). The transport time scales are broadly consistent with previous estimates. For example, using a trajectory model coupled with tracer emissions, Stohl (2006) showed that, of the air parcels from Asia transported to the Arctic in winter, 25% arrived in less than 14 days and 50% in more than 20 days. Harrigan et al. (2011) attributed BC sampled along an April 12, 2008 DC-8 flight during the ARCTAS field campaign (Jacob et al., 2010) to Asian emissions 10–15 days prior to sampling times. At Denali, Barrow, Alert, and Zeppelin, all four low-level sites, model results show persistent influences from Asian anthropogenic emissions of BC, emitted as far back as March 6–16, after journeys of 40–50 days of transport, presumably after being transported to the Arctic circle and entrained by and becoming part of the circumpolar transport around and within the Arctic circle (Ma et al., 2013). These transport time scales are born out by the ~2-month lifetime of Asian anthropogenic emissions of BC against deposition in the Arctic (Table 1). Direct, episodic transport of Asian emissions during April 8−15 account for 66% of the total Asian contribution (from emissions during March 1–April 25) to BC at Denali and 37% at Barrow. In contrast, Asian emissions during March 6−16 and the subsequent circumpolar transport (hence 'chronic' pollution, Brock et al., 2011) account for 39% of the total Asian contribution to BC at Barrow, 65% at Alert and 57% at Zeppelin. These results suggest that both direct, episodic events and chronic Asian anthropogenic pollution, on time scales of 1–2 months, play comparable roles in enhancing springtime Arctic surface BC. Previous studies identifying sources of Arctic surface BC used 5- or 10-day back trajectory analyses (Pollisar et al., 2001; Sharma et al., 2004; Eleftheriadis et al., 2009; Huang et al., 2010a; Matsui et al., 2011; Dutkiewicz et al., 2014). As such, the relatively short time scales used in those studies likely result in significant underestimates of long-range transport of BC from Asia to the Arctic.





At Denali, North American emissions have nearly an immediate impact, on a timescale of 0–9 days, because of the proximity to regional sources (power plants and petroleum refinery industries) in Alaska. These episodic transport events are the major parts of North American contributions at Denali (98%) and to a much lesser degree at Barrow (64%). There are secondary maxima in the North American anthropogenic contributions during March 16−31 (a time lag of 20–30 days) at

Denali, Barrow. These maxima reflect circumpolar transport of North American sources from Canada and the lower 48 states, hence the longer time lags. At Alert and Zeppelin, no direct transport events from North American anthropogenic sources are seen. Instead, circumpolar transport dominates the contribution from North American anthropogenic sources, which account for ~9% of total BC at Alert and ~5% at Zeppelin. At Denali, Barrow, Alert and Zeppelin, there is no direct transport from European anthropogenic emissions during this period. Rather, a long tail in March (more than 25 days of time

lag) is seen at each of the four sites, most evident at Barrow, Alert and Zeppelin. These are European anthropogenic emissions that had been circulating the Arctic troposphere. Their relative contributions are 10% at Barrow, 18% at Alert and 13% at Zeppelin. Siberian anthropogenic contributions, mostly from natural gas flares (42% at Alert and 62% at Zeppelin) in the WENR (Stohl et al., 2013; Qi et al., 2016a), have large impacts on BC at Alert and Zeppelin after 6–15 days of transport. Gas flaring contributions are relatively small at Denali (~3%) and Barrow (~7%) and negligible at Summit (~1%) because of

the weak sensitivities (Sect. 4.2.1).

Overall, episodic, direct transport dominates anthropogenic contributions at Denali (out of Arctic front, 87%), while chronic, circumpolar transport dominates anthropogenic contributions at Alert (furthest North in the Arctic of the five sites, 89%). The two types of contributions are comparable at Barrow (direct: 42%) and Zeppelin (direct: 52%). BC concentration from

direct transport events at Barrow (12.9 ng m$^{-3}$) is more than one order of magnitude lower than that at Denali (177.4 ng m$^{-3}$). This difference is largely explained by the strong transport barrier, i.e., the 'Arctic front' at ~66°N (Stohl, 2006 and references therein), which lies between and separates Denali (63.7°N) and Barrow (71.3°N). BC concentration at 750 hPa, above the Arctic front, over Barrow is more than a factor of three larger than that at the surface during April 20–25 (Fig. 7). Contributions from both Asian anthropogenic sources and Siberian biomass burning emissions to BC at 750 hPa Barrow

(Asian anthropogenic: 57.0 ng g$^{-1}$; Siberian biomass burning: 119.8 ng g$^{-1}$) are four times larger than those at the surface Barrow (Asian anthropogenic: 13.6 ng g$^{-1}$; Siberian biomass burning: 30.4 ng g$^{-1}$) and are comparable to their contributions to Denali (Asian anthropogenic: 38.3 ng g$^{-1}$; Siberian biomass burning: 137.8 ng g$^{-1}$). The Arctic front also traps BC emitted within the front to the Arctic lower troposphere. This trapping is evident at Barrow – North American anthropogenic emissions are significant sources for BC at the surface (13%) but negligible at 750 hPa (1%). Zeppelin is strongly influenced

by direct transport of natural gas flaring emissions in the WENR and European anthropogenic emissions. BC at Alert is also enhanced by direct transport from these flaring emissions, but to a much less extent (a factor of five lower) compared with





Zeppelin. In contrast, almost all biomass burning contributions (> 97%) are from direct transport events in April at the four sites.

BC at Summit shows remarkably different signatures of source types and source regions than that at the other four sites. This free tropospheric site experiences a large and persistent contribution from anthropogenic sources of BC emitted during March 1–April 15 in Asia, spanning time scales of 10–50 days. North American anthropogenic emissions take as short as four days (4–16 days) to impact BC at Summit because of the close proximity. Somewhat surprisingly, Summit is subject to comparable influences from European and North American anthropogenic emissions (~20%). Anthropogenic BC emitted during April 8–15 in Europe arrives at the site after 10–17 days of transport.

### 4.3 Uncertainty analysis

### 4.3.1 Uncertainty associated with the temporal resolution of biomass burning emissions

Open biomass burning emissions are known to have large day-to-day and diurnal variations (Giglio, 2007). It is conceivable that the temporal resolution of biomass burning emissions used in the model may introduce significant uncertainty in our source attribution. Here we examine the sensitivity of BC at the five sites to temporal variation of Siberian biomass burning emissions and explore the implications of these variations on atmospheric transport of BC to the Arctic. Specifically, we use both a monthly and a 3-hourly biomass burning emission inventories (Sect. 3.1) to assess the abovementioned uncertainty. The left panels of Fig. 8 show both monthly and 3-hourly BC biomass burning emissions in Siberia in March–April 2008. Also overlaid is the adjoint sensitivity of Arctic surface BC (at the five sites) to the emissions. Biomass burning contributions to Arctic surface BC are thus the combined results of the spatiotemporal distribution of the emissions and the sensitivities (Sect. 3.2). Fig. 8 (right panels) shows the resulting contributions from Siberian biomass burning to mean BC concentrations on April 20−25 at the five sites. The 3-hourly inventory results in more significant day-to-day and diurnal variations, as expected, and more acute episodic contributions (e.g., at Barrow from emissions on April 9–10 and 13–15). In contrast, the monthly inventory leads to more broad and persistent contributions from emissions on April 1–15, with more mesoscale variations. The temporal variation of contributions in April estimated from the monthly inventory generally follows the variation of sensitivities. Contribution maxima correspond to sensitivity maxima, such as the contribution peaks on April 7–10 at Barrow and on April 8–11 at Alert. Contributions estimated using 3-hourly inventory also peak on the same days. Contributions in March are negligible because of very low biomass burning emissions. Overall, the 3-hourly inventory leads to weaker polar-ward transport of BC. For instance, the contribution is 50% lower at Denali, 31−39% lower at Barrow, Alert and Zeppelin, and 6% lower at Summit. The lower contributions are likely because the temporal variation of the 3-hourly inventory is out of phase of the sensitivities at all sites (Fig. 8 left panels).





### 4.3.2 Uncertainty associated with wet scavenging

We examine here the role of wet scavenging, a determining factor of BC loading in the Arctic (Huang et al., 2010b; Vignati et al., 2010; Liu et al., 2011; Browse et al., 2012; Qi et al., 2016), on polar-ward transport of BC. A critical process that affects the wet scavenging of BC particles is WBF (Cozic et al., 2007; Henning et al., 2004). In a previous study, we have

shown that WBF releases BC particles incorporated in cloud water drops back into interstitial air in mixed-phase clouds, thereby strongly reduces BC scavenging efficiency and slows down subsequent wet scavenging (Qi et al., 2016b). Conversely, the absence of WBF leads to lower mean BC concentrations in surface air. To examine the effect of WBF on Arctic surface BC, we conduct a simulation whereby WBF is turned off. A direct consequence of the absence of WBF is weaker sensitivities of Arctic BC to global emissions. Fig. 9 shows the reductions of these sensitivities relative to the

standard simulation (which includes WBF). In the absence of WBF, the sensitivities are lower by 10–90% for most regions. The resulting BC concentrations in surface air at the five sites are lower by 18–52%. The negative discrepancy against observations is further exaggerated at Barrow and Alert.

The absence of WBF results in inhomogeneous reduction of sensitivities. For Denali, Barrow, Alert and Zeppelin, the

reductions of sensitivities are larger in far field source regions and smaller to near field emissions. The resulting relative contributions are higher from near field, but lower from far field. At Denali for example, the WBF effect results in significantly larger reductions of sensitivities to emissions from Europe (50–70%) than to those from the North American sector in the Arctic and the North Pacific (20–40%). Consequently, the resulting relative contribution from North America are larger in the absence of WBF (50%) than that with WBF included (43%). Similarly, for Barrow the relative contribution

increases from 12% to 17% for North American emissions and from 49% to 51% for Siberian emissions in the absence of WBF, while the relative contributions decrease for other sources. For Alert and Zeppelin, the relative contributions from proximate source region Siberia increase from 46% to 53% at Alert and from 66% to 72% at Zeppelin. This indicates that WBF strongly increases the polar-ward transport of BC from far field regions. For Summit, the reduction of sensitivity without WBF is more evenly distributed than other surface sites and the resulting relative contributions from different

sources change marginally.

### 5. Summary and conclusions

This study identified sources of BC at surface sites in the low Arctic (Denali), the high Arctic (Barrow, Alert and Zeppelin) in the Arctic free troposphere (Summit) using a 3D global chemical transport model GEOS-Chem with concentrations tagged by emission source regions at the continental scale in April 2008. We also identified sources and temporal variations





of BC during a pollution episode at these five sites at 2° latitude × 2.5° longitude horizontal and hourly temporal resolution using the GEOS-Chem adjoint model.

The tagger tracer technique showed that the largest sources of BC in April 2008 were Asian anthropogenic sources

(35–45%) and Siberian biomass burning emissions (46–64%). Adjoint sensitivity showed that during a transport episode in April 20−25, BC at Denali and Barrow in Alaska was most sensitive to emissions in the Pacific ocean while BC at Alert and Zeppelin was most sensitive to emissions in the Arctic Circle and Eurasia. At Summit in the free troposphere, BC was more sensitive to emissions at lower than higher altitudes.

The fine horizontal resolution of adjoint helps identify forest fires in southern Siberia-Lake Baikal area as the largest sources of biomass burning, accounting for more than 90% of total global biomass burning contributions. The largest anthropogenic sources of BC at Denali and Barrow were local and regional emissions in Alaska. In addition, residential emissions from Heilongjiang province in Northeast China was a large anthropogenic source for Denali. For Alert and Zeppelin, the largest anthropogenic sources were gas flares in WENR. Anthropogenic sources (industrial and residential) in Jing-jin-ji megacities

and Shandong province in China were large sources (~10%) of BC at Barrow, Alert and Zeppelin in high Arctic.

The transport time scales can be inferred from the temporal contribution spectra. Except for the immediate impact of local sources in Alaska to Denali and Barrow, other sources took more than 5 days to reach the surface sites. It took six days for Siberian anthropogenic emissions to reach Zeppelin and Alert. Biomass burning emissions in South Siberia and Asian

anthropogenic emissions took more than 10 days of transport to influence BC concentration at surface Arctic. The fine temporal resolution of contribution differentiated the contributions of episodic, direct transport from that of chronic, circumpolar transport. Our results suggested that even during this strong pollution episode, direct transport accounted for half of the contribution at most (42% at Barrow, 52% at Zeppelin and 11% at Alert). The chronic, circumpolar transport of BC is the largest contributor to BC at surface. A large fraction of the Asian contribution was from the chronic circumpolar

transport (~60% at Barrow and ~100% at Alert and Zeppelin). The long direct transport time scale (>12 days) and the chronic transport dominated contribution suggested that contribution from Asia was strongly underestimated by previous studies that were based on 5- or 10-day trajectory analysis.

Source attribution of Arctic surface BC using the adjoint method was associated with uncertainties from all processes,

including emissions, transport, aging, and deposition. We found that using a 3-hourly temporal resolution of biomass burning emissions reduced BC concentration strongly at Denali (~50%), moderately (~30%) at Barrow, Alert and Zeppelin and marginally (6%) at Summit. Finer resolution did not affect global sensitivity. The decrease of contribution resulted from the



emissions and sensitivities being out of phase. BC concentrations at the five sites were lower by 35–52% without WBF, resulting from a reduction of sensitivities to global emissions. Without WBF, the sensitivity decreased more in the far field sources faraway than near field sources for Denali, Barrow, Alert and Zeppelin. Thus, the relative contribution from proximate sources increased in the absence of WBF. This indicates that WBF strongly increases the polar-ward transport

from far field regions. For Summit, the change of the relative contribution from different regions without WBF was negligible.

## Acknowledgements

This study was funded by NASA grant NNX14AF11G from the Atmospheric Chemistry Modeling and Analysis Program

(ACMAP) and by a grant from the U.S. Environmental Protection Agency's Science to Achieve Results (STAR) program. Although the research described in the article has been funded completely or in part by the U.S. Environmental Protection Agency's STAR program through grant R835037, it has not been subjected to any EPA review and therefore does not necessarily reflect the views of the Agency, and no official endorsement should be inferred. We would also like to thank the Global Monitoring Division at NOAA Earth System Research Laboratory, the Atmospheric Science and Technology

Directorate at Environment Canada and SFT Norway for providing data. The Swedish Environmental Protection Agency and the Swedish Research Council have sponsored BC measurements at Zeppelin.

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





**Table 1: Atmospheric lifetimes of Arctic BC (2008) from major sources and source regions (days).**

|  |  | Annual | March-April |
|---|---|---|---|
| Anthropogenic | North America | 22 | 36 |
|  | Europe | 9 | 16 |
|  | Siberia | 7 | 10 |
|  | Asia | 47 | 71 |
| Biomass Burning | Siberia | 16 | 30 |
|  | South Asia | 98 | 141 |



**Table 2: Model simulated relative contributions to surface BC concentration (ng m$^{-3}$) at selected Arctic sites (Fig. 1) for April 20–25[*], 2008.**

| | | Anthropogenic | | | | Biomass burning | | Total | Adjoint |
|---|---|---|---|---|---|---|---|---|---|
| | | Asia | Siberia | Europe | N.A. | Siberia | S. Asia | | Forward |
| Denali | Forward | 41.9 | 10.4 | 11.1 | 142.4 | 142.4 | 0.33 | 352.1 | |
| | Adjoint | 38.3 | 9.5 | 10.0 | 146.0 | 137.8 | 0.26 | 341.9 | 97% |
| Barrow | Forward | 15.0 | 4.4 | 5.9 | 7.6 | 27.8 | 0.15 | 60.1 | |
| | Adjoint | 13.6 | 4.3 | 5.5 | 7.4 | 30.4 | 0.11 | 61.8 | 103% |
| Alert | Forward | 16.6 | 10.1 | 10.8 | 4.7 | 14.3 | 0.25 | 58.5 | |
| | Adjoint | 13.8 | 9.5 | 9.3 | 4.5 | 14.3 | 0.15 | 51.8 | 89% |
| Zeppelin | Forward | 21.0 | 32.8 | 14.6 | 5.7 | 39.3 | 0.35 | 116.5 | |
| | Adjoint | 17.8 | 32.4 | 13.4 | 5.3 | 39.3 | 0.23 | 108.8 | 93% |
| Summit | Forward | 17.2 | 0.8 | 7.5 | 6.1 | 5.9 | 1.15 | 42.0 | |
| | Adjoint | 14.3 | 0.7 | 7.1 | 7.2 | 4.8 | 1.13 | 38.0 | 90% |

[*]Values are averages for April 20–25. 'Forward' and 'Adjoint' are from 'tagged tracer' and from adjoint simulations. See text for details.





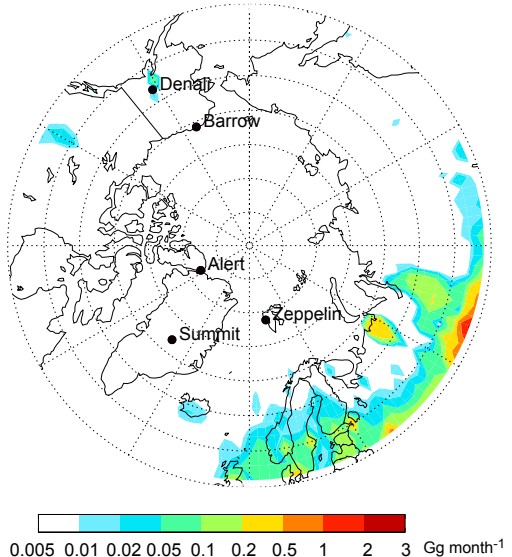

**Figure 1: BC emissions (Gg month$^{-1}$) in the Arctic for April. Solid circles are surface sites: Denali, AL (63.7°N, 149.0°W, 0.66 km a.s.l.), Barrow, AL (71.3°N, 156.6°W, 0.01 km a.s.l.), Alert, Canada (82.3°N, 62.3°W, 0.21 km a.s.l.), Summit, Greenland (72.6°N, 38.5°W, 3.22 km a.s.l.), and Zeppelin, Norway (79°N, 12°E, 0.47 km a.s.l.). Data from Bond et al. (2007) and natural gas flaring emissions from Stohl et al. (2013).**





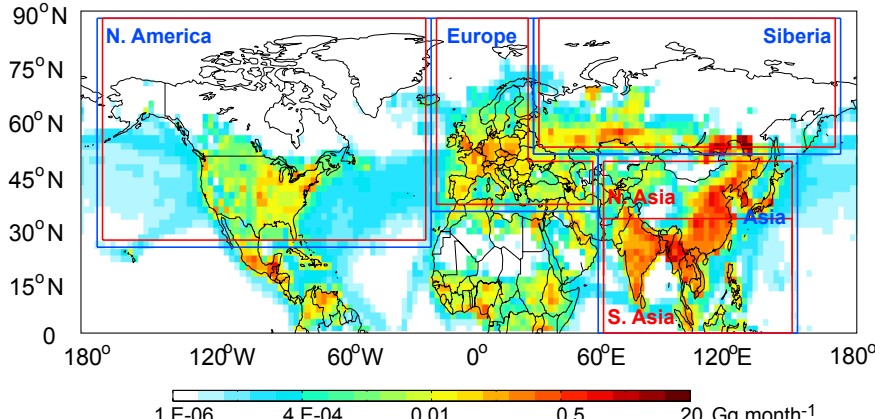

**Figure 2: Northern Hemispheric BC emissions for April (Gg month[-1]). BC tracers are 'tagged' by four anthropogenic (blue rectangles) and five biomass burning sources (red rectangles): North America (172.5–17.5°W, 24–88°N) – both anthropogenic and biomass burning, Europe (17.5°W–30°E, 33–88°N and 30–60°E, 33–50°N) – both, Siberia (30–172.5°E, 50–88°N) – both, Asia – anthropogenic (60–152.5°E, 0–50°N), Northern China (60–152.5°E, 30–50°N) – biomass burning sources, South Asia and Southern China (60–152.5°E, 0–30°N) – biomass burning.**





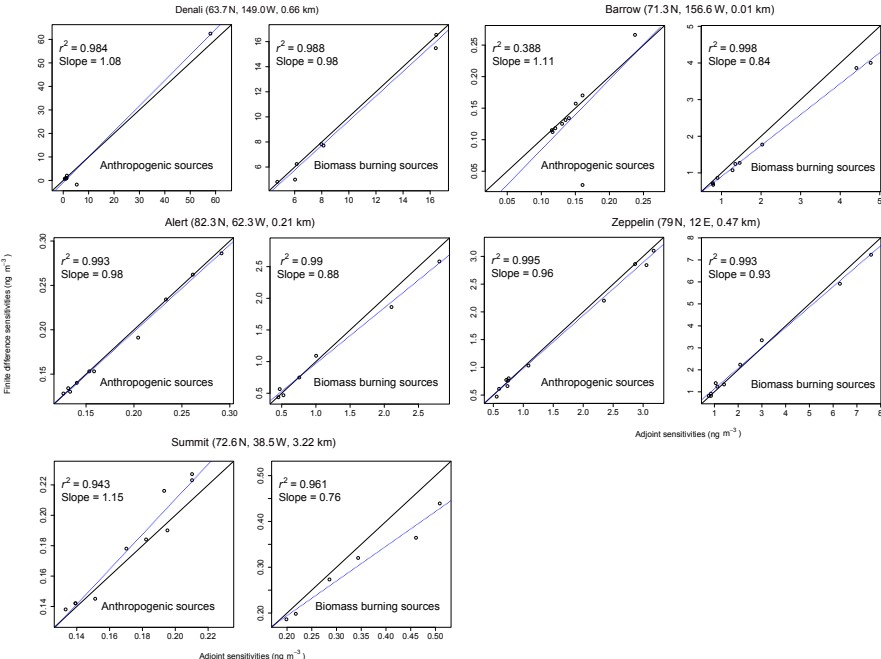

**Figure 3:** GEOS-Chem adjoint versus finite difference gradients of BC concentration for 10 model gridcells with largest anthropogenic BC contributions and 10 gridcells with largest biomass burning BC contributions at Denali, Barrow, Alert, Zeppelin, and Summit (see Fig. 1). Values are for March 1–April 25, 2008. Regression lines, slopes and $r^2$ values are shown.





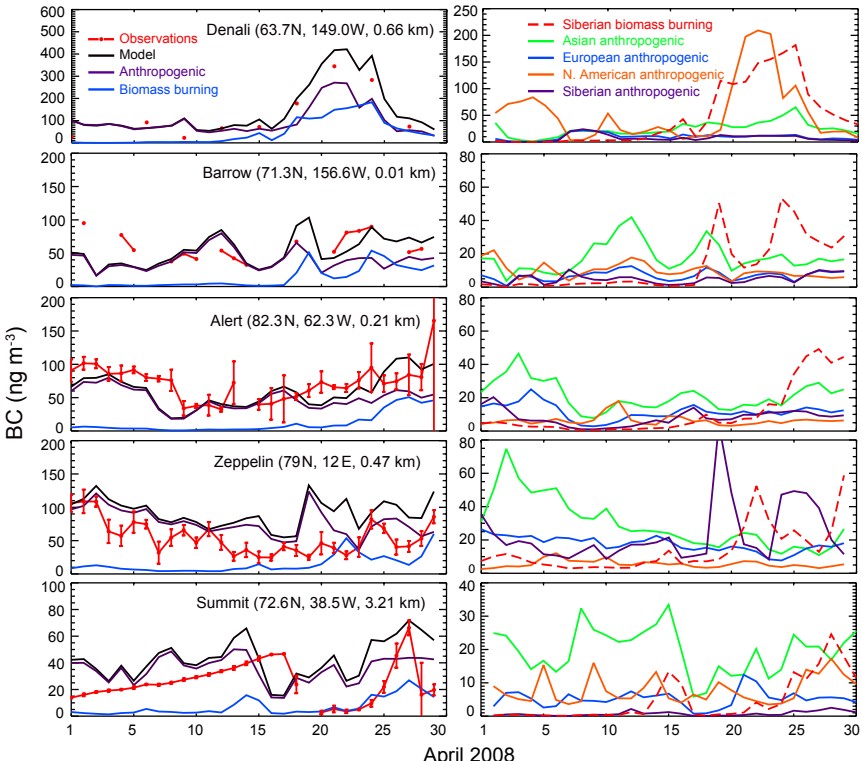

**Figure 4: (left panels) Observed (red lines) and GEOS-Chem simulated (black lines) daily mean BC concentrations at Denali, Barrow, Alert, Zeppelin and Summit (see Fig. 1) for April 2008. Bars are standard deviations of observations. Modeled BC concentrations are decomposed into contributions from anthropogenic (purple lines) and biomass burning emissions (blue lines). (right panels) major contributions: Siberian biomass burning (red dashed line), Asian anthropogenic (green solid line), European anthropogenic (blue solid line), North American anthropogenic (orange solid line), and Siberian anthropogenic (purple solid line) sources.**





**Figure 5: Sensitivity of mean BC concentration (averages for April 20–25, 2008) at Denali, Barrow, Alert, Zeppelin, and Summit (see Fig. 1) to Northern Hemispheric BC emissions from during the 5, 10 and 25 days prior to April 25. See text for details.**





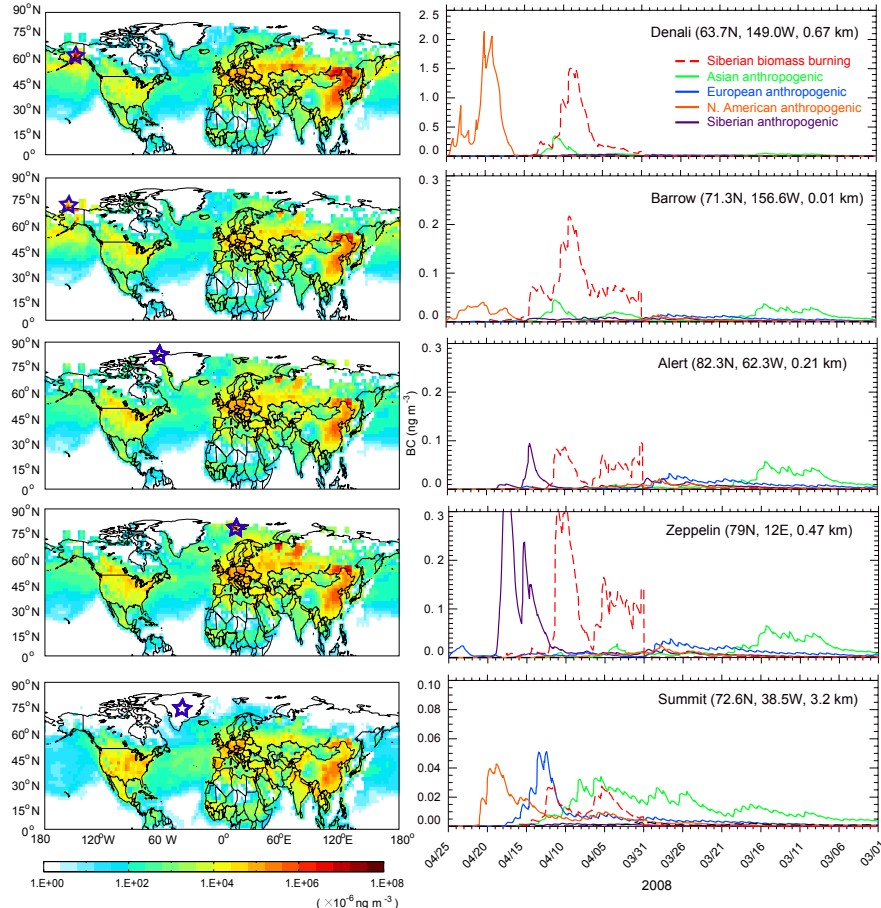

**Figure 6: GEOS-Chem simulated contributions to BC concentrations, averaged over April 20–25, 2008, at (stars, left panels) Denali, AL (63.7°N, 149.0°W, 0.66 km a.s.l.), Barrow, AL (71.3°N, 156.6°W, 10 m a.s.l.), Alert, Canada (82.3°N, 62.3°W, 210 m a.s.l.), Zeppelin, Norway (79°N, 12°E, 470 m a.s.l.), and Summit, Greenland (72.6°N, 38.5°W, 3.22 km a.s.l.) from Northern hemispheric emissions as computed from adjoint simulations (see Sect. 3.2 for details): (left panels) Cumulative contributions from March 1–April 25 and (right panels) time-dependent contributions from Siberian biomass burning (red dashed line), Asian anthropogenic (solid green line), European anthropogenic (solid blue), North American anthropogenic (solid orange), and Siberian anthropogenic sources (solid purple). Source regions are as defined in Fig. 2.**





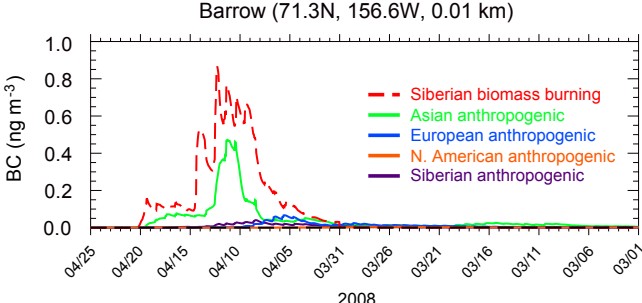

**Figure 7: Same as Fig. 6 (right panels) but for BC at 750 hPa above Barrow.**





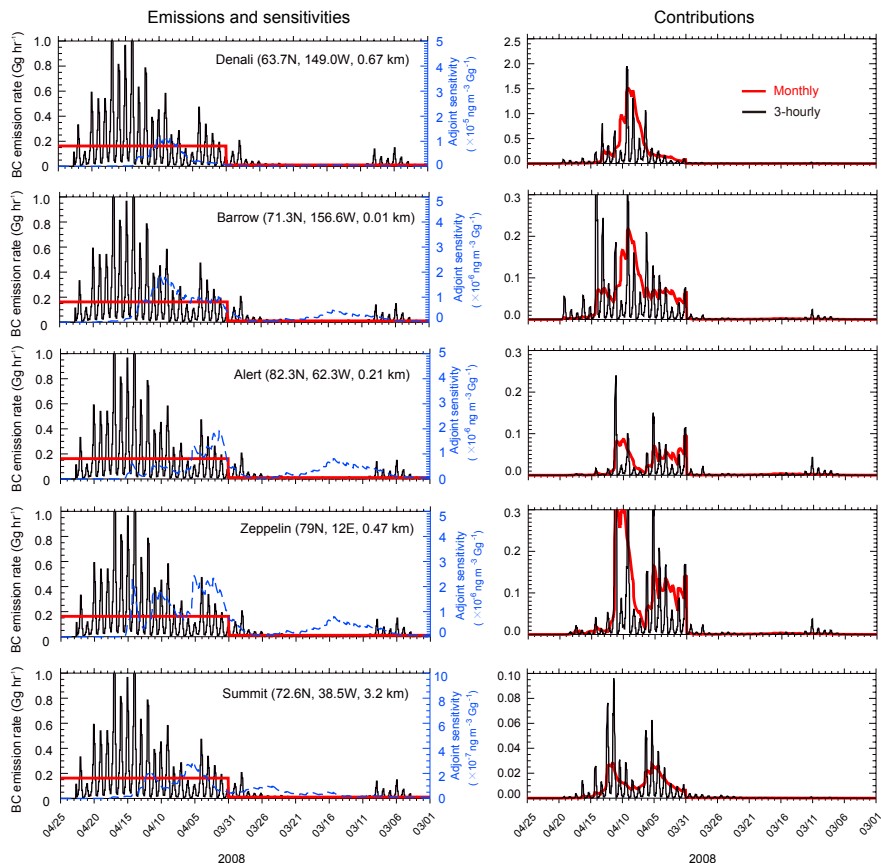

**Figure 8: (left) Biomass burning BC emission rates (Gg hr⁻¹, red –monthly, black – 3-hourly) in Siberia (see Fig. 2) for April 2008 and (right) contributions to BC at the five sites from Siberian biomass burning emissions (red – monthly, black – 3-hourly) during March 1–April 25. Emission data is from the Global Fire Emissions Database version 3 (GFEDv3) inventory (Randerson et al., 2012). Adjoint sensitivities of mean BC during April 20–25, 2008 at the five sites (see Fig. 1) to Siberia biomass burning emissions are also shown (left panel, dashed blue).**





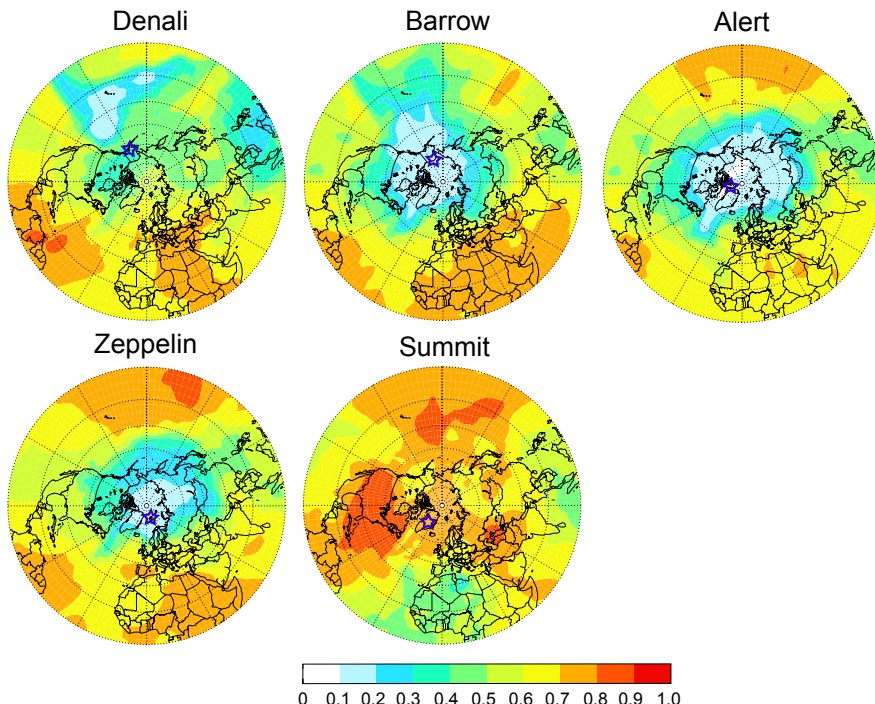

**Figure 9: Reduction of adjoint sensitivities of BC concentration at selected Arctic sites (Fig. 1) to Northern Hemispheric emissions in the absence of WBF relative to standard simulation, averaged for the whole simulation period in March 1–April 25, 2008.**