# Peer review of "Sources of Springtime Surface Black Carbon in the Arctic: An Adjoint Analysis for April 2008"

_Atmospheric Chemistry and Physics, 2016_

## Referee Comment (RC1) · Anonymous Referee #1 · 20 Mar 2017

This is an interesting study exploring sources of black carbon to the Arctic during April 2008, using an adjoint model configuration. The paper is well-written and well-referenced. Overall I find this to be a suitable study for publication in ACP and a worthwhile contribution to the literature in general, but I include below some critiques that should be addressed before the study is published.

Major comments:

The authors raise the important point that previous studies applying 5- or 10-day back trajectories probably underestimate the contribution of BC from Asia to the Arctic. Yet the authors here cite surprisingly long atmospheric lifetimes for some sources of BC (Table 1), including 71 days for springtime Asia anthropogenic emissions, 98 days for annual-mean South Asia biomass emissions, and 141 days for springtime South Asia

biomass emissions. Since the model runs were initiated only on March 1, less than two months before the analysis period, the long atmospheric lifetimes of springtime Asia emissions imply that Asia contributions may also have been underestimated in this study. The authors should include some assessment of the potential magnitude of bias introduced by their short spinup period compared with the modeled BC atmospheric lifetime.

More generally, the reported lifetimes (Table 1) seem substantially longer than those cited in previous studies, and imply that the simulated global atmospheric burden (and probably direct radiative forcing) of BC must be quite large in this model. Please comment on this, in the context of previous studies of global BC burdens and radiative forcing.

While it is fine to focus the study on a narrow timeframe (April 2008), there is large interannual variability in spring BC emissions, especially those associated with biomass burning, and probably also in transport pathways to the Arctic. I suggest acknowledging this a bit more clearly, and if possible briefly discussing how representative the conditions of April 2008 were of Aprils in general (say, over the period 2000-2015). I also suggest changing the title from "Sources of Springtime..." to "Sources of Springtime 2008..." to communicate that this study only quantitatively examines one particular spring (unless the authors expand the analysis in a revised draft).

The modeled biases with respect to observations are as large as 80% at some locations. The implications of such large biases for source apportionment should be described more thoroughly. It is unlikely that all sources are biased in the same proportion, implying the potential for considerable bias in the apportionment itself.

Minor comments:

p1,16: Do these percentages represent the fraction of anthropogenic sources or the fraction of total BC? Wording suggests the former, but please clarify.

p2,2: Please clarify the spatial domains over which these sensitivity numbers apply. Presumably the temperature change is Arctic, but over what domain is forcing averaged over?

p3,3: "ends" -> "end"

p3, second paragraph: This section highlights great features of the adjoint technique and provides good justification for the methods applied here, but I would also acknowledge clearly that the quality of an adjoint analysis still depends on the accuracy of the physical representations built into the forward version of the model.

p5,21: Was the gridded flaring emissions inventory used in this study provided completely by Stohl et al (2013), or were additional assumptions adopted to create the inventory? Please clarify.

p5: Were any shipping emissions included in this study? If not, please comment on this omission and its potential importance for the study.

p5,26-31: It wasn't clear to me which aging scheme was applied in the main analysis of this paper. For the main analysis, was constant aging assumed or slower winter/spring aging assumed? Please clarify.

p5,31: "We estimates" -> "We estimate"

p6,31: "reaching the sites": Presumably you mean reaching the surface (i.e., lowest model layer) at these locations, but please clarify.

p7,2: "entirely linear": linear with respect to what? Please clarify.

p7,25: their -> there

p8,13-14: But would this feature necessarily produce a positive bias, as seen in the analysis?

p8,28: This is one place where the representativeness of 2008 could be briefly described (see major comment).

p9,30+: Please clarify whether these lifetimes are Arctic or global lifetimes, and describe how they were calculated. Also see comment below about Table 1.

p10,28: "in the Arctic front" is used here and elsewhere. The front itself is a boundary, so it might be more clear to instead use something like "within the polar dome" or "poleward of the Arctic front".

p11,10: just for clarity I suggest changing "mean BC concentrations" to "mean forward-simulated BC concentrations".

p12,22: meaning of "against deposition in the Arctic" is unclear to me.

p13,5: "Denali, Barrow" -> "Denali and Barrow" (?)

p14,27: "Overall, the 3-hourly inventory leads to weaker polar-ward transport of BC. For instance, the contribution is 50% lower at Denali,..." - Do these changes improve or worsen the agreement with measurements? Presumably, they should improve the comparison, no? Related:

p14,29-30: "The lower contributions are likely because the temporal variation of the 3-hourly inventory is out of phase of the sensitivities at all sites" - But this should be a physically realistic phenomenon since the sensitivities were derived from high temporal resolution re-analysis data, shouldn't it? Perhaps the importance of this passage could be clarified by re-phrasing it in terms of physical processes rather than model sensitivities. Related:

And finally, why were monthly emissions used for the main analysis? It seems that 3-hourly emissions should produce more realistic assessments.

p15,6: "thereby" -> "and thereby"

p16,4: "sources of BC" -> "sources of BC to the Arctic"

[Figure]

p16,11: "contributions": contributions to what? Arctic atmospheric BC? Please specify.

Table 1: Again, please clarify exactly what these "lifetimes" represent. Are they residence times of BC within the Arctic? If so, how were they computed? If a parcel of BC enters, leaves, and re-enters the Arctic, how would this affect the "lifetime"? Or are these global lifetimes of BC that reaches the Arctic?

Figure 5: Maybe clarify that these are sensitivities with respect to hypothetical unit emissions occurring everywhere.

Figure 8: Are the sensitivities for 3-hourly or monthly resolved emissions? What are the units of the right panel?

Figure 9: What are the units? Please describe.

---

## Referee Comment (RC2) · Anonymous Referee #3 · 28 Mar 2017

Qi et al. quantify black carbon (BC) emission sources at 5 stations in the Arctic during April 2008 using a global CTM; GEOS-Chem and the adjoint technique. The authors use a tagged tracer technique to identify the sources, and compare the contributions with observations at the sites. A 5-day pollution episode is investigated further by computing source-receptor sensitivities at finer resolution with adjoint modeling approach. They find that the largest sources during April 2008 are anthropogenic emissions from Asia and biomass burning emissions from Siberia.

Arctic bC burdens is mostly a result of long-range transport, and determining the emissions source regions is challenging. This study is therefore an important contribution to the field and within the scope of ACP. The Introduction chapter is well-written with a great overview of related work, and the authors clearly indicate their own contribution describing the GEOS-Chem adjoint approach. The overall presentation is wellstructured and easy to follow. I will highly recommend this manuscript for publication and I have a few comments, questions, and suggestions for improvements below.

1. The validity of your results depends on the emission data set. Can you please add more information about the emissions? What are the global numbers of the Bond et al. (2007) emissions combined with the Asian emissions from Zhang Q. et al. (2009)? Can you also give the total emissions from each region (those numbers can be added to Table 1)? On page 5 in Line 20 you state that Stohl et al suggests that gas flaring only accounts for 3 % of global emissions, but 42 % of the within-Arctic. Can't you report on your own emissions numbers here? Do you use ECLIPSE (v5?) emissions for the gas flaring?

2. As you state in the introduction, many studies based on analysis of observation data, attribute Arctic BC to biomass burning and anthropogenic sources in high-latitude Eurasia (e.g., Eleftheriadis et al., 2009; Hirdman et al., 2010; Matsui et al., 2011). Compared to observation-based analysis in the Arctic, models, including yours, tend to give larger contributions to sources at lower latitudes, especially for the total column burden of Arctic BC. However, models often have too course resolution to correctly simulate the Arctic front and the shallow boundary layer. Could you please add some discussion of uncertainty in simulating the transport in GEOS-Chem?

3. You compare your results nicely with other findings throughout the Results section. Could you also add a paragraph in your Conclusions with a summary of the comparison of your findings with previous studies to wrap things up? -If possible, why they differ (e.g. the 5-10 day trajectory used in other studies that you mention in the intro), and how your study contribute to greater knowledge of BC source attribution in the Arctic?

4. You report a long BC lifetime from Asian BB emissions. How realistic do you think this number is? As far as I am aware, this is considerably higher than other studies.

5. I suggest that you add 'April 2008' in your title, as readers might assume you have analyzed several years/months (as I did). Related to this; can you add some discussion

if April 2008 is representative of 'springtime'? E.g. the biomass burning plume late April 2008 was unusually strong (Warneke et al. 2010).

P5 L5: What are the lowest model levels listed related to?

P10 L28: In –> within?

P15 L11: What about the 3 other stations? You have already showed that the model overestimate the BC concentrations at Zeppelin and Summit? Is there a better agreement without WBF?

P15 L27: , after Zeppelin) .. add ' and,'

P16 L8: altitudes –> latitudes?

---

## Author Comment (AC1) · 26 May 2017

**Referee #1**

**Major Comments:**

*"This is an interesting study exploring sources of black carbon to the Arctic during April 2008, using an adjoint model configuration. The paper is well-written and well-referenced. Overall I find this to be a suitable study for publication in ACP and a worthwhile contribution to the literature in general, but I include below some critiques that should be addressed before the study is published."*

1 *"The authors raise the important point that previous studies applying 5- or 10-day back trajectories probably underestimate the contribution of BC from Asia to the Arctic. Yet the authors here cite surprisingly long atmospheric lifetimes for some sources of BC (Table 1), including 71 days for springtime Asia anthropogenic emissions, 98 days for annual-mean South Asia biomass emissions, and 141 days for springtime South Asia biomass emissions. Since the model runs were initiated only on March 1, less than two months before the analysis period, the long atmospheric lifetimes of springtime Asia emissions imply that Asia contributions may also have been underestimated in this study. The authors should include some assessment of the potential magnitude of bias introduced by their short spinup period compared with the modeled BC atmospheric lifetime."*

**Response**: Points well taken. The long lifetime reported previously in the manuscript only accounted for deposition. I revised the lifetime now to account for both deposition and transport (Table 1). The method was explained in the note of Table 1. The lifetimes for BC accounting for both deposition and transport are less than 12 days. Thus, the two-month spin up time before the starting date March 1 is long enough for the system.

2 *"More generally, the reported lifetimes (Table 1) seem substantially longer than those cited in previous studies, and imply that the simulated global atmospheric burden (and probably direct radiative forcing) of BC must be quite large in this model. Please comment on this, in the context of previous studies of global BC burdens and radiative forcing."*

**Response**: Points well taken. As explained in Response for question #1, the revised BC lifetime in the Arctic is close to other studies now. Global BC burden is within the range of current AeroCom models (See details in Table 6 in our papers Qi et al., 2017a, b).

3 *"While it is fine to focus the study on a narrow timeframe (April 2008), there is large inter-annual variability in spring BC emissions, especially those associated with biomass burning, and probably also in transport pathways to the Arctic. I suggest acknowledging this a bit more clearly, and if possible briefly discussing how representative the conditions of April 2008 were of Aprils in general (say, over the period 2000-2015). I also suggest changing the title from "Sources of Springtime..." to "Sources of Springtime*

*2008..." to communicate that this study only quantitatively examines one particular spring (unless the authors expand the analysis in a revised draft)."*

**Response**: Points well taken. Revised accordingly in the title and in Sect. 3.1.

4 *"The modeled biases with respect to observations are as large as 80% at some locations. The implications of such large biases for source apportionment should be described more thoroughly. It is unlikely that all sources are biased in the same proportion, implying the potential for considerable bias in the apportionment itself."*

**Response**: Points well taken. Revised accordingly in Sect. 4.3.3.

**Specific comments:**

1. *"p1,16: Do these percentages represent the fraction of anthropogenic sources or the fraction of total BC? Wording suggests the former, but please clarify."*

**Response**: It is the former. Clarified.

2. *"p2,2: Please clarify the spatial domains over which these sensitivity numbers apply. Presumably the temperature change is Arctic, but over what domain is forcing averaged over?"*

**Response**: Clarified.

3. *"p3,3: "ends" -> "end""*

**Response**: Done.

4. *"p3, second paragraph: This section highlights great features of the adjoint technique and provides good justification for the methods applied here, but I would also acknowledge clearly that the quality of an adjoint analysis still depends on the accuracy of the physical representations built into the forward version of the model."*

**Response**: Points well taken. Acknowledged as such in Page 4 Lines 1–2.

5. *"p5,21: Was the gridded flaring emissions inventory used in this study provided completely by Stohl et al (2013), or were additional assumptions adopted to create the inventory? Please clarify."*

**Response**: The flaring emission inventory used in this study is completely from Stohl et al. (2013). Clarified in Sect. 3.1.

6. *"p5: Were any shipping emissions included in this study? If not, please comment on this omission and its potential importance for the study. "*

**Response**: Anthropogenic emissions used in this study (Bond et al., 2007) include

shipping emissions already.

7. *"p5,26-31: It wasn't clear to me which aging scheme was applied in the main analysis of this paper. For the main analysis, was constant aging assumed or slower winter/spring aging assumed? Please clarify."*

**Response**: Clarified.

8. *"p5,31: "We estimates" -> "We estimate""*

**Response**: Done.

9. *"p6,31: "reaching the sites": Presumably you mean reaching the surface (i.e., lowest model layer) at these locations, but please clarify."*

**Response**: Clarified accordingly.

10. *"p7,2:"entirely linear": linear with respect to what? Please clarify."*

**Response**: Linear with respect to emission. Clarified accordingly.

11. *"p7,25: their -> there "*

**Response**: Done.

12. *"p8,13-14: But would this feature necessarily produce a positive bias, as seen in the analysis?"*

**Response**: Clarified.

13. *"p8,28: This is one place where the representativeness of 2008 could be briefly described (see major comment)."*

**Response**: Revised. See response for question #3.

14. *"p9,30+: Please clarify whether these lifetimes are Arctic or global lifetimes, and describe how they were calculated. Also see comment below about Table 1."*

**Response**: Revised accordingly in the note of Table 1.

15. *"p10,28: "in the Arctic front" is used here and elsewhere. The front itself is a boundary, so it might be more clear to instead use something like "within the polar dome" or "poleward of the Arctic front"."*

**Response**: Points well taken. Revised in the manuscript.

16. *"p11,10: just for clarity I suggest changing "mean BC concentrations" to "mean forward simulated BC concentrations"."*

**Response**: Done.

17. *"p12,22: meaning of "against deposition in the Arctic" is unclear to me."*

**Response**: Explained in the noted of Table 1.

18. *"p13,5: "Denali, Barrow" -> "Denali and Barrow" (?)"*

**Response**: Done.

19. *"p14,27: "Overall, the 3-hourly inventory leads to weaker polar-ward transport of BC. For instance, the contribution is 50% lower at Denali,..." - Do these changes improve or worsen the agreement with measurements? Presumably, they should improve the comparison, no? "*

**Response**: Clarified.

20. *"p14,29-30: "The lower contributions are likely because the temporal variation of the 3-hourly inventory is out of phase of the sensitivities at all sites" - But this should be a physically realistic phenomenon since the sensitivities were derived from high temporal resolution re-analysis data, shouldn't it? Perhaps the importance of this passage could be clarified by re-phrasing it in terms of physical processes rather than model sensitivities.*

**Response**: Revised accordingly.

21. *And finally, why were monthly emissions used for the main analysis? It seems that 3-hourly emissions should produce more realistic assessments."*

**Response**: Explained in Sect. 3.1.

22. *"p15,6: "thereby" -> "and thereby""*

**Response**: Done.

23. *"p16,4: "sources of BC" -> "sources of BC to the Arctic""*

**Response**: Done.

24. *"p16,11: "contributions": contributions to what? Arctic atmospheric BC? Please specify.*

**Response**: Clarified.

25. *"Table 1: Again, please clarify exactly what these "lifetimes" represent. Are they residence times of BC within the Arctic? If so, how were they computed? If a parcel of BC enters, leaves, and re-enters the Arctic, how would this affect the "lifetime"? Or are these global lifetimes of BC that reaches the Arctic?" "*

**Response**: Clarified in notes of Table 1.

26. *"Figure 5: Maybe clarify that these are sensitivities with respect to hypothetical unit emissions occurring everywhere."*

**Response**: Clarified.

27. *"Figure 8: Are the sensitivities for 3-hourly or monthly resolved emissions? What are the units of the right panel?"*

**Response**: They are the contributions (ng m$^{-3}$) from Siberia biomass burning emissions with 3-hourly and monthly resolutions, respectively. Figure 8 is revised.

28. *"Figure 9: What are the units? Please describe."*

**Response**: It is unitless. The figure plots the ratio of sensitivities without and with the WBF effect.

**References**
Bond, T. C., Bhardwaj, E., Dong, R., Jogani, R., Jung, S. K., Roden, C., Streets, D. G., and Trautmann, N. M.: Historical emissions of black and organic carbon aerosol from energy-related combustion, 1850-2000, Global Biogeochem Cy, 21, doi 10.1029/2006gb002840, 2007.
Qi, L., Li, Q., Li, Y., and He, C.: Factors Controlling Black Carbon Distribution in the Arctic, Atmos. Chem. Phys., 17, 1037-1059, doi:10.5194/acp-17-1037-2017, 2017a.
Qi, L., Li, Q., He, C., Wang, X., and Huang, J.: Effects of Wegener-Bergeron-Findeisen Process on Global Black Carbon Distribution, Atmos. Chem. Phys., in press, 2017b.

---

## Author Comment (AC3) · 26 May 2017

**Referee #3**

**Major Comments:**

*"Qi et al. quantify black carbon (BC) emission sources at 5 stations in the Arctic during April 2008 using a global CTM; GEOS-Chem and the adjoint technique. The authors use a tagged tracer technique to identify the sources, and compare the contributions with observations at the sites. A 5-day pollution episode is investigated further by computing source-receptor sensitivities at finer resolution with adjoint modeling approach. They find that the largest sources during April 2008 are anthropogenic emissions from Asia and biomass burning emissions from Siberia.*

*Arctic BC burden is mostly a result of long-range transport, and determining the emissions source regions is challenging. This study is therefore an important contribution to the field and within the scope of ACP. The Introduction chapter is well-written with a great overview of related work, and the authors clearly indicate their own contribution describing the GEOS-Chem adjoint approach. The overall presentation is well-structured and easy to follow. I will highly recommend this manuscript for publication and I have a few comments, questions, and suggestions for improvements below."*

1 *"The validity of your results depends on the emission data set. Can you please add more information about the emissions? What are the global numbers of the Bond et al. (2007) emissions combined with the Asian emissions from Zhang Q. et al. (2009)? Can you also give the total emissions from each region (those numbers can be added to Table 1)? On page 5 in Line 20 you state that Stohl et al suggests that gas flaring only accounts for 3 % of global emissions, but 42 % of the within-Arctic. Can't you report on your own emissions numbers here? Do you use ECLIPSE (v5?) emissions for the gas flaring?"*

**Response**: Points well taken. Revised accordingly in Sect. 3.1 and Table 1.

2 *"As you state in the introduction, many studies based on analysis of observation data, attribute Arctic BC to biomass burning and anthropogenic sources in high-latitude Eurasia (e.g., Eleftheriadis et al., 2009; Hirdman et al., 2010; Matsui et al., 2011). Compared to observation-based analysis in the Arctic, models, including yours, tend to give larger contributions to sources at lower latitudes, especially for the total column burden of Arctic BC. However, models often have too course resolution to correctly simulate the Arctic front and the shallow boundary layer. Could you please add some discussion of uncertainty in simulating the transport in GEOS-Chem?"*

**Response**: Excellent point. Added discussion in Sect. 4.3.3.

3 *"You compare your results nicely with other findings throughout the Results section. Could you also add a paragraph in your Conclusions with a summary of the comparison of your findings with previous studies to wrap things up? -If possible, why they differ (e.g.*

*the 5-10 day trajectory used in other studies that you mention in the intro), and how your study contribute to greater knowledge of BC source attribution in the Arctic?"*

**Response**: Points well taken. Revised in Sect. 5.

4  *"You report a long BC lifetime from Asian BB emissions. How realistic do you think this number is? As far as I am aware, this is considerably higher than other studies."*

**Response**: Points well taken. The lifetime I reported in the manuscript was against deposition only. I revised the lifetime considering both transport and deposition in Table 1. This lifetime is close to the other studies.

5  *"5. I suggest that you add 'April 2008' in your title, as readers might assume you have analyzed several years/months (as I did). Related to this; can you add some discussion if April 2008 is representative of 'springtime'? E.g. the biomass burning plume late April 2008 was unusually strong (Warneke et al. 2010)."*

**Response**: Points well taken. Revised accordingly in the title and in Sect. 3.1.

**Specific comments:**

1. *"P5 L5: What are the lowest model levels listed related to?"*

**Response**: We only discuss the surface sites in this study, so we deleted this description.

2. *"P10 L28: In –> within?"*

**Response**: Done.

3. *"P15 L11: What about the 3 other stations? You have already showed that the model overestimates the BC concentrations at Zeppelin and Summit? Is there a better agreement without WBF?"*

**Response**: Clarified as suggested.

4. *"P15 L27: , after Zeppelin) .. add ' and,'"*

**Response**: Done.

5. *"P16 L8: altitudes –> latitudes?"*

**Response**: Done.

[revised manuscript text omitted]